# Antibacterial Activities of Homemade Matrices Miming Essential Oils Compared to Commercial Ones

**DOI:** 10.3390/antibiotics10050584

**Published:** 2021-05-14

**Authors:** Sofia Oliveira Ribeiro, Véronique Fontaine, Véronique Mathieu, Zhiri Abdesselam, Baudoux Dominique, Stévigny Caroline, Souard Florence

**Affiliations:** 1Department of Research in Drug Development (RD3), Pharmacognosy, Bioanalysis and Drug Discovery Unit, Faculty of Pharmacy, Université Libre de Bruxelles, Boulevard du Triomphe, 1050 Brussels, Belgium; Caroline.Stevigny@ulb.be; 2Department of Research in Drug Development (RD3), Microbiology, Bioorganic and Macromolecular Chemistry Unit, Faculty of Pharmacy, Université Libre de Bruxelles (ULB), Boulevard du Triomphe, 1050 Brussels, Belgium; veronique.fontaine@ulb.be; 3Department of Pharmacotherapy and Pharmaceutics, Université Libre de Bruxelles (ULB), Boulevard du Triomphe, 1050 Brussels, Belgium; Veronique.Mathieu@ulb.ac.be; 4Pranarôm International S.A. 37, Avenue des Artisans, 7822 Ghislenghien, Belgium; abdesselam.zhiri@ulb.ac.be (Z.A.); dbaudoux@pranarom.com (B.D.); 5Unité de Recherche en Biotechnologie Végétale, Université Libre de Bruxelles, CP 300, Rue Prof. Jeener & Brachet 12, 6041 Gosselies, Belgium; 6Department of Pharmacotherapy and Pharmaceutics (DPP), Pharmacology, Pharmacotherapy and Pharmaceutical Care Unit, Faculty of Pharmacy, Université Libre de Bruxelles (ULB), Boulevard du Triomphe, 1050 Brussels, Belgium; Florence.Souard@ulb.be; 7Département de Pharmacochimie Moléculaire (DPM), Université Grenoble Alpes, CNRS, UMR 5063, F3Y041 Grenoble, France

**Keywords:** Thymol, *β*-caryophyllene, ajowan essential oil, thyme essential oil, antibacterial activity, *Staphylococcus aureus*

## Abstract

The increasing bacterial resistance to antibiotics is a worldwide concern. Essential oils are known to possess remarkable antibacterial properties, but their high chemical variability complicates their development into new antibacterial agents. Therefore, the main purpose of this study was to standardize their chemical composition. Several commercial essential oils of ajowan (*Trachyspermum ammi* L.) and thyme (chemotype thymol) (*Thymus vulgaris* L.) were bought on the market. GC–MS analysis revealed that thyme essential oils have a chemical composition far more consistent than ajowan essential oils. Sometimes thymol was not even the major compound. The most abundant compounds and the homemade mixtures were tested against two *Staphylococcus aureus* strains. The antibacterial property of *β*-caryophyllene presented no direct activity against *S. aureus* LMG 15975, but in association with thymol or carvacrol at equal percentages an MIC of 125 μg/mL was observed. The mixture of those three compounds at equivalent percentages also decreased by 16-fold the MIC of the penicillin V. Against *S. aureus* LMG 21674, *β*-caryophyllene presented an MIC of 31.3 μg/mL and decreased by 267-fold the MIC of the penicillin V. These observations led us to question the benefits of using a complex chemical mixture instead of one active compound to fight bacterial resistance.

## 1. Introduction

With the progress of science in the modern world and especially since the discovery of penicillin in 1928 by Fleming [1], it seemed that the use of natural products was no longer an essential need. However, as a result of the antibiotic resistance crisis and the general population seeking natural treatments, research on natural products and notoriously on essential oils is back on the rise [2,3,4,5]. Essential oils are generally composed of hundreds of secondary metabolites that confer a multitude of biological activities [3,6]. Despite the antibacterial potential of essential oils and the amount of literature describing their biological properties, clinical studies are scarce with most being related to mouthwashes, and actual use of essential oils by healthcare professionals still seems a distant goal [7]. This could be due to the fact that their clinical safety is sometimes questionable [8] and/or they possess a chemical composition that can easily vary. Indeed, their chemical composition depends on both biotic or abiotic factors and a batch can be chemically different over the course of a year even if it comes from the same harvested place and the same producer [6,9]. The aim of this study was to find a way to standardize the chemical composition of essential oils by preparing a homemade mixture composed of the most abundant compounds since only two or three major compounds are often accountable for the antibacterial activities [10]. Ajowan (*Trachyspermum ammi* L.) and thyme chemotype thymol (*Thymus vulgaris* L.) essential oils were chosen due to their close chemical composition, both having thymol as a major compound, a monoterpenoid phenol known to possess antibacterial properties [11,12]. The use of thymol as an additive to polymers in wound skin infection treatment has been recently investigated [13,14]. This point is interesting since the aim in this paper was to investigate the homemade mixtures against the Gram-positive Staphylococcus aureus, which is considered as a high threat to human health by the World Health Organization [15] and often present in skin infections [16].

## 2. Materials and Methods

### 2.1. Essential Oils, Biological and Chemical Material

Eight ajowan essential oils and eight thyme chemotype thymol essential oils were purchased online or in pharmacies in order to have a large panel of suppliers. Table 1 lists information relative to the purchased products available for consumers, such as the supplier, the origin and the plant organ used, the batch number, and the price per liter. *Staphylococcus aureus* LMG 15,975 and *S. aureus* LMG 21,674 were purchased from the Belgian Coordinated Collection of Microorganisms (BCCM/LMG, Ghent, Belgium). Keratinocyte cell line (HaCaT) was purchased from Promocell (Germany). All chemical materials, solvents, bacterial mediums, 3-(4,5-dimethylthiazol-2-yl)-2,5-diphenyltetrazolium bromide (MTT), and reference compounds were purchased from Sigma-Aldrich (Saint-Louis, MO, USA). Physiological solution (NaCl 0.85%, 2 mL) was purchased from BioMérieux (Marcy-l’Étoile, France). Antibiotics were purchased from Alfa Aesar (Ward Hill, MA, USA). Fetal bovine serum was purchased from Greiner (Kremsmünster, Autriche). Cell growth medium and supplements were purchased from Thermofisher (Waltham, MA, USA).

### 2.2. Preparation of Homemade Mixtures

The mixtures were prepared at the desired percentages and diluted in DMSO (5% maximum) to a final concentration of 2 mg/mL (same concentration for the essential oils and for compounds alone). Mixture 1 (M1) was an ajowan mixture comprised of 50% thymol, 25% *γ*-terpinene, 20% *p*-cymene, and 5% *β*-pinene. Mixture 2 (M2) was a thyme chemotype thymol mixture comprised of 50% thymol, 20% *p*-cymene, 10% *γ*-terpinene, and 20% other minor compounds (5% each: *β*-myrcene, linalool, *β*-caryophyllene, and carvacrol). Mixtures 3 to 7 (M3 to M7) contained an increasing concentration of thymol and the same decreasing concentrations of *γ*-terpinene and *p*-cymene. Mixture 8 to 10 (M8 to M10) were mixtures two by two (50/50 *v*/*v*) of the three most active compounds, thymol, carvacrol, and *β*-caryophyllene, and mixture 11 (M11) was a mixture 33.3/33.3/33.3 (*v*/*v*/*v*) of the three compounds together. GC–MS analyses were performed on these mixtures to verify if the concentrations were well respected since the mixtures were made according to the volume and not the molecular weight of each compound.

### 2.3. GC–MS Analysis

All the essential oils and the mixtures were analyzed via gas chromatography—GC (Trace GC ultra, Thermo, Waltham, MA, USA)—coupled with a mass spectrometer—MS (Polaris-Q, Thermo, Waltham, MA, USA)—in order to identify the main chemical compounds. An Rxi-5Sil MS (Restek, France) (40 m × 0.18 mm; thickness 0.18 µm) column was used and helium was the carrier gas (1 mL/min). A solution (0.8 µL) prepared in hexane at 0.5% was injected for each essential oil (splitless mode) with a temperature program set as follows: 50 °C for 6 min followed by an increase of 2 °C/min until reaching a temperature of 250 °C, which was then held for 10 min. The detector and injector temperatures were fixed at 200 °C and 240 °C, respectively. Major compounds were first analyzed in order to identify their retention indices, which were further used to identify the essential oils’ components. Peak area percentages were used to obtain quantitative data (*n* = 3).

### 2.4. Cytotoxic Activity

The effect of each essential oil on the global growth of the cell line was investigated through a colorimetric MTT assay [17] as previously described [18]. Briefly, human keratinocytes cells (HaCaT) were grown in RPMI supplemented with 10% fetal bovine serum and 0.5% penicillin-streptomycin and seeded at a density of 10 000 cells/mL in 96-well plates. After 24h of incubation at 37 °C in a 5% CO_2_ humidified atmosphere, the cells were treated with three concentrations of the essential oils (31.3, 62.5, and 125 μg/mL). The contact time of incubation was 72 h. Finally, after 3 to 5 hours of incubation with the MTT (0.5 mg/mL), the plates were centrifuged at 1200 rpm for 5 min, the supernatant was discarded, and the formazan crystals formed dissolved in DMSO. A spectrophotometer (680XR, BioRad, Hercules, CA, USA) was used to measure the absorbance (at 570 and 630 nm), which can be directly correlated to the number of viable cells (cell viability (% RC) = (absorbance of treated cells/absorbance of control) × 100).

### 2.5. Antibacterial Activity

Two types of antibacterial assays were employed as previously described [18]. All the samples, essential oils, mixtures of essential oil types, and individual compounds were evaluated alone (direct activity) or in association with an antibiotic (indirect activity) at a sub-MIC concentration (non-active concentration). The tested strains were the Gram-positive *Staphylococcus aureus* LMG 15,975 (methicillin-resistant *S. aureus*, abbreviated as MRSA) and *S. aureus* LMG 21,674 (fusidic-acid-resistant *S. aureus*, abbreviated as FUSR). The chosen antibiotics were penicillin V and fusidic acid. The broth microdilution method in 96-well plates was used to determine the minimum inhibitory concentration (MIC). For the direct activity, samples were serially diluted to achieve a concentration ranging from 1 mg/mL to 2 μg/mL. For the indirect activity, the antibiotic was first serially diluted (from 64 to 7.103 μg/mL) and the samples were subsequently added to the wells at the defined sub-MIC concentration. These ranges were chosen because above them the sample is considered as non-active. For both assays, the inoculum was prepared from a suspension of 24-hour colonies to obtain a turbidity of 0.5 McFarland and added to the wells to achieve a final concentration of 5 × 10^5^ CFU/mL. Plates were then incubated at 37 °C overnight and MTT (0.8 mg/mL) was used to determine the MIC by naked eye. A growth (Mueller–Hinton broth medium with inoculum) and a non-growth (only MHB) control were carried out. In addition, antibiotics were also always tested alone as positive controls.

### 2.6. Statistical Analysis

All experiments were performed, at least, in triplicate. Median values were calculated from all obtained antibacterial independent measurements. Regarding the GC–MS analysis and the cytotoxic activity, the presented data are expressed as mean values ± standard deviation (SD).

## 3. Results and Discussion

Essential oils are known to possess antibacterial properties, especially against Gram-positive bacteria, such as *Staphylococcus aureus*, a skin infection pathogen [7,16] considered by the World Health Organization as a high threat to human health [15]. Since essential oils can have variable chemical compositions [19] and their antibacterial activity generally comes from their major compounds [10], homemade mixtures of ajowan and thyme (chemotype thymol) essential oils containing only the major compounds were prepared. Eight ajowan essential oils and eight thyme chemotype thymol essential oils were bought on the market and their chemical compositions were analyzed via GC–MS. The antibacterial activities of the commercial essential oils, homemade mixtures, and a selection of chemical compounds were investigated individually (direct activity) or in combination (indirect activity) against two *S. aureus*-resistant strains. For the indirect activity, the investigated products were combined at a sub-MIC concentration with penicillin V and fusidic acid. Those *β*-lactam and steroidal antibiotics are commonly used to treat either systemic or topical MRSA infections, respectively, although MRSA are usually more resistant to both of them [1,20,21].

Table 1 shows information about the purchased ajowan and thyme chemotype thymol essential oils. The terms used by the supplier for the plant organ were maintained. In the case of the ajowan essential oils, the terms fruit and seed refer to the same organ, but the correct botanical identification is fruit. The last column shows the price at which the essential oils were bought. The price is obviously not indicative of effectiveness since many factors (from producing to marketing) could influence it. It was considered opportune to mention the price to illustrate that price is not always related to the quality of the product. The chemical composition, antibacterial activity, geographic origin, and “organic” status were compared. All the ajowan essential oils, from eight different suppliers, were native to India. Three of them were listed as organic (European and French labels), and regarding the price, they were valued under the medium market price of this essential oil (51 ± 22 €/L).

The most expensive (92.2 €/L) was the essential oil from Pranarôm (AW-S1) purchased in a drugstore and the cheapest (28.5 €/L) was the one purchased online from Voshuiles (AW-S3). Regarding the thyme chemotype thymol essential oil, six different suppliers sold it with a mean price of 109 ± 51 €/L. Among the eight thyme chemotype thymol essential oils, half were coming from Spain and the other half from France. The difference of the mean price between the two origins was quite large, 69 ± 17 €/L (Spain) and 148 ± 41 €/L (France). Most of the information relative to the plant (scientific name, plant organ, and origin) were mentioned on the package, except for the sample AW-S6 from Floressence where the producing organ was not mentioned. Those and other pieces of information, such as, safety uses, instructions for use, expiration dates or batch identifications, should always be mentioned on the package. Not just for the purpose of the scientists who study them but mostly as a guarantee of quality for consumers [22].

*Thymus vulgaris* L. essential oil possesses different chemotypes [23]. The investigated ones in this study had thymol as the major compound. The thyme essential oil is described in the European pharmacopeia as an essential oil obtained by steam distillation of the aerial flowering fresh parts of *Thymus vulgaris* L., *T. zygis* L., or a mixture of both [24]. The major compounds of thyme chemotype thymol essential oil are well defined by ranges in the European pharmacopeia, and regarding the ones bought in the market for this study, their chemical composition did not fluctuate between the suppliers (Table 2). The percentage means obtained for their major compounds via GC–MS analysis showed that they were approximatively in the ranges indicated by the European pharmacopeia (between parentheses in the text): with 57% (37–55%) for thymol, 26% (14–28%) for *p*-cymene, and 8% (4–12%) for *γ*-terpinene.

Being a chemotype essential oil as described by the European pharmacopoeia seems to force suppliers to pay more attention to the composition of their batches. In fact, ajowan essential oil is not described in the European pharmacopeia, and related or not, its chemical composition tends to be irregular among suppliers (Table 3). Only AW-S6, and quite comparably AW-S1, had a composition with thymol as the major compound, followed by *p*-cymene and *γ*-terpinene present at equivalent percentages (around 20%). Among the other purchased essential oils, some had *γ*-terpinene as the major compound (AW-S2 and AW-S3) or in percentage equivalent to thymol (AW-S4) while others had *p*-cymene (AW-S7) as the major compound or in percentage equivalent to thymol (AW-S5 and AW-S8). However, thymol as the major compound (around 50%) of ajowan essential oil is stated as being the normal composition [25,26]. As mentioned before, it is known that the composition of an essential oil can vary among the same species because of many factors, such as genetic profiles, environmental changes, harvesting parameters, mode of obtention, and so on [19]. If this variability is scientifically noteworthy, it should not be acceptable to detect such high variability in the composition of commercially available essential oils obtained from plant materials characterized by a similar botanical definition. Essential oils could be used as therapeutic products. It should be a priority for the consumer to be informed about the exact major composition of the essential oils since a different composition can lead to a different biological activity than the one stated by the manufacturer.

Table 4 shows the biological activities of the purchased ajowan and thyme chemotype thymol essential oils (AW-S1 to S8 and TT-S1 to S8), of the individual compounds and of the mixtures (M1 to M11).

After 72 hours of contact, all the samples presented a cytotoxicity on the HaCaT cells at 125 μg/mL (mean residual viability under 80%), but only the linalool was toxic at all tested concentrations, even at the smallest (31.3 μg/mL). According to the interpretation proposed by Döll-Boscardin et al. [27], linalool is the only compound with strong cytotoxic activity (IC_50_ < 31.3 μg/mL), all the others presented moderate (31.3 μg/mL < IC_50_ < 125 μg/mL) to weak (IC_50_ > 125 μg/mL) cytotoxic activity. Accordingly, to avoid a potential cytotoxicity, the sub-MIC concentrations selected for the indirect antibacterial activity were under the observed cytotoxic threshold for each sample.

The indirect antibacterial activity of the samples in association with fusidic acid are not presented in Table 4 since an improved indirect antibacterial activity with this antibiotic could not be detected. Fusidic acid is principally active against Staphylococci, but the emergence of resistant strains is increasing [20]. The breakpoint to consider a Staphylococci resistance is 1 mg/L [28]. In this study, a specific *S. aureus* resistant to fusidic acid (FUSR) with a tested MIC at 256 μg/mL (*S. aureus* LMG 21674) was used. However, no indirect activity was observed, and the MIC of the tested association remained at the same MIC as the one of the antibiotic alone. To our knowledge, there is no literature about the positive association of fusidic acid with an essential oil against a resistant strain of *S. aureus*.

Concerning the direct activity toward *S. aureus* LMG 21,674 (FUSR), the ajowan essential oils presented an MIC of 250 μg/mL, and the thyme chemotype thymol essential oil had an MIC of 125 μg/mL. Against the other MRSA strain (LMG 15975), the essential oils were less active, with a MIC two-fold superior. Despite the variability in chemical compositions of the ajowan essential oils, there was no difference observed in the direct activity against those two strains. In order to identify the compound(s) responsible for the activity, they were tested alone. No direct activity against any of the tested strains were presented by *p*-cymene and *γ*-terpinene. It seems that thymol is the main compound responsible for the direct activity. The same MICs were observed for thymol and thyme chemotype thymol essential oils comprised of at least 50% thymol. For the ajowan essential oils, the thymol was present in lesser concentrations and consequently the MIC of ajowan essential oils were two-fold superior to the ones of thymol on both strains.

As mentioned by many authors, due to the high variability of the test methods, the adjustments made by the authors, and the variability in chemical compositions of the tested essential oils, comparison of the results between studies is very difficult [16,29,30,31]. For ajowan essential oil, no studies were found using the same tested strains. However, against the MRSA ATCC 700698, Zomorodian et al. [32] found an MIC of 1 mg/mL, close to the 500 μg/mL observed for the MRSA LMG 15795, but 4-fold higher than the one found for the LMG 21,674 strain. In the case of thyme essential oil, against the same strain as in this study, Boskovic et al. [33] found an MIC of 320 μg/mL and Sienkiewicz et al. [34] found an MIC of 25 μg/mL with thyme oils composed of 50.48% and 38.1% thymol, respectively. These results are quite similar to the ones obtained in this study where the samples had an average thymol concentration of 56.92%. Essential oils containing thymol are often described as possessing potent antibacterial properties [35]. Regarding the antibacterial activity of thymol against MRSA strains, results found in the literature are often comparable to the ones obtained in this study. Kifer et al. [36] and Hamoud et al. [37] found MICs of 125 to 250 μg/mL against strains identified as MRSA. Nostro et al. [38] and Cirino et al. [39] found MICs a little bit higher, around 600 μg/mL, against the same strain and other MRSA strains, respectively. Of note, those concentrations are toxic against human immortalized keratinocytes (Table 4).

The indirect activity presented in this study made use of a different method to test synergistic effects between two compounds. The difference with the traditional checkerboard method was that compound A (antibiotic) was serially diluted and compound B (essential oil/pure metabolite or homemade mixture) was added at a fix sub-MIC concentration, determined under the cytotoxic activity threshold. This method allowed us to observe directly the increased activity of the antibiotic by the diminution of its MIC. An indirect activity is observed when the MIC of the association is lower than the MIC of the antibiotic alone. Concerning the commercial essential oils, the conclusions drawn from the indirect activity were quite similar to the ones from direct activity. No indirect activity was presented by *p*-cymene and *γ*-terpinene on both strains. However, thymol decreased four-fold the MIC of penicillin V on FUSR *S. aureus* (16 to 4 μg/mL) and on MRSA (4 to 1 μg/mL). The indirect activity of thymol was two-fold higher than the one of some essential oils, which can be explained by its concentration, present around 50% in the essential oils. But for some purchased essential oils there was no activity observed with the penicillin V. The hypothesis is that some antagonisms happened or that it was the effect of the variable chemical composition. This observation led us to the central point of the study: could isolated active compounds or a blend of the most active molecules of essential oils generate better antibacterial activities than genuine commercial products?

To answer this question, the biological activities of the major compounds and some minor compounds present in the ajowan and thyme chemotype thymol essential oils were first tested. Apart from thymol, two compounds showed a direct activity. Carvacrol presented an MIC of 250 μg/mL against the strain MRSA and an MIC of 62.5 μg/mL against the FUSR *S. aureus* strain. Still on the FUSR *S. aureus*, *β*-caryophyllene showed an MIC of 31.3 μg/mL. The MIC observed for carvacrol and *β*-caryophyllene are quite interesting since they are active at concentrations identified as non-cytotoxic in the tested cell line, HaCaT. Carvacrol is known to possess antibacterial properties against a large variety of strains [3,40]. On the contrary, *β*-caryophyllene is rarely referenced in the literature for its antibacterial activity. This sesquiterpene has been classified as safe to be used as a flavor ingredient in fragrances and in cosmetics by many authorities [41]. Some studies have demonstrated it has different biological properties, such as antifungal [42,43], antileishmanial [44], and antitrypanosomal properties [45,46]. Lately, studies have been focusing on other qualities, such as its anti-inflammatory [47], antidiabetic [48], and anticancer and chemopreventive properties [41,43]. Nevertheless, fewer studies refer to the antibacterial properties of *β*-caryophyllene. Dahham et al. (2015) found a better MIC on *S. aureus* MTCC 7405 than the kanamycin. No indication about the resistance of the strain MTCC 7405 was found. As part of his study on balsam fir, Pichette et al. [49] found an MIC of 5.1 μg/mL against *S. aureus* ATCC 25923. The difference with the MIC found in this study (31.3 μg/mL) is not surprising since the strain ATCC 25,923 is an MSSA. By contrast, DeCarlo et al. [50] found an MIC of 313 μg/mL against an MSSA (ATCC 29,213). As mentioned above, even if the same microdilution method was used different strains and adaptations made to the protocol can lead to different results.

The indirect activity of *β*-caryophyllene is more impressive than its direct activity. Against the FUSR *S. aureus*, the concentration of penicillin V was decreased by 267-fold (16 to 0.06 μg/mL) when associated with *β*-caryophyllene at a sub-MIC of 15 μg/mL, a concentration under the observed toxicity threshold on HaCaT. Even if it is less impressive, this compound, at a non-cytotoxic sub-MIC, also decreased by 4-fold the concentration of penicillin V on the MRSA. Carvacrol and thymol, at a sub-MIC of 32 μg/mL, also presented good indirect activity with a 4-fold decreased MIC on FUSR *S. aureus* and a 4-fold to 8-fold decrease on MRSA, respectively. Carvacrol and thymol are known to be synergistic with different antibiotics against different pathogens, including penicillin on *S. aureus* [39,51,52]. To our knowledge, this is the first time that a positive interaction between *β*-caryophyllene and an antibiotic against *S. aureus* has been observed.

The aim of this study was to obtain an efficient antibacterial “homemade mixture-type essential oil” composed of only major compounds of ajowan and thyme chemotype thymol essential oils. The mixture-type ajowan (M1) with 50% thymol, 25% *γ*-terpinene, 20% *p*-cymene, and 5% *β*-pinene presented the same level of activity as ajowan essential oils. Similar statements can be made for the mixture type thyme chemotype thymol (M2) with 50% thymol, 20% *p*-cymene, 10% *γ*-terpinene, 5% *β*-myrcene, 5% linalool, 5% *β*-caryophyllene, and 5% carvacrol. As for the mixtures M3 to M7, where thymol was increased and *p*-cymene/*γ*-terpinene decreased, it was observed that these last two compounds did not enhance thymol activity. As the thymol was increased and the two others were decreased, only the full activity of the thymol, as if it were present alone, was measured. As the cytotoxicity of thymol was similar (less than 62.5 μg/mL) to ajowan and thyme chemotype thymol essential oils and the prepared mixtures, there was, in those cases, no interest in using them over thymol alone. However, the advantage of the prepared mixtures over the ajowan essential oils was that their composition was controlled and consequently their antibacterial activity would not depend on the batch.

Given the interesting properties of thymol, carvacrol, and *β*-caryophyllene, other mixtures (M8 to M11) where tested with different concentrations. *β*-caryophyllene was a little bit more cytotoxic against the HaCaT cell line than the two other compounds. Therefore, the mixtures containing *β*-caryophyllene (M9, M10, and M11) presented a cytotoxicity above 31.3 μg/mL while M8, containing 50% thymol and 50% carvacrol, presented a cytotoxicity above 62.5 μg/mL. The cytotoxic level of compounds can vary between cell lines, but it is very important to always verify that the antibacterial activity is not related to a high level of toxicity. Very few papers checked the cytotoxic activity on normal cell lines as a control. Most of the publications regarding the cytotoxic effects of essential oils or their compounds were focused on their anticancer activity. Among those that tested healthy cells, thymol and carvacrol presented low cytotoxicity on human lung cells [53] and *β*-caryophyllene showed a low cytotoxic effect on green monkey kidney, mouse fibroblast, human retinal ganglion, and human colon fibroblast cell lines [43,45]. These results are in line with the ones described in this study where the IC_50_ was located between 32 and 125 μg/mL, which represents a moderate to weak cytotoxicity. The advantage to using controlled homemade mixtures or compounds alone is that negative effects, as cytotoxicity, can be more standardly regulated. The balance of benefit and risk can be more stable compared to essential oils whose composition can vary considerably.

Regarding antibacterial activity, no enhancement of the activity was observed for the thymol/carvacrol mixture (M8), for both direct and indirect activities, against any of the strains. In fact, the activity of the compounds alone was equivalent or better. Similar results and even some antagonisms were found by Rúa et al. [54] when testing the carvacrol/thymol combination against different strains of *S. aureus*. The same observation as for M8 was made for the direct activity of M9, M10, and M11 against FUSR *S. aureus*, where the MIC was similar to the one observed for *β*-caryophyllene alone. Therefore, it seems just as beneficial to use the compound alone since the cytotoxicity is the same. However, against the strain MRSA, the results of mixtures M9, M10, and M11 were quite interesting. It seems that *β*-caryophyllene, which did not present a direct activity (>1000 μg/mL), enhances the activity of either thymol or carvacrol when mixed together in equal parts (M9 and M10, respectively) and when it is mixed with thymol and carvacrol in equal proportions (M11). Those MIC values of 125 μg/mL were the best ones observed against the strain *S. aureus* LMG 15975.

The indirect activities of these three mixtures in association with penicillin V are also enhanced compared to the indirect activities of the compounds alone. However, the activity of *β*-caryophyllene alone against FUSR *S. aureus* (0.06 μg/mL) was still the best observed. Even if the activity of carvacrol and thymol were quite enhanced, from 4 μg/mL to 0.5 μg/mL and 0.125 μg/mL, respectively, it would be best to use *β*-caryophyllene alone in this particular case with penicillin V and against the strain *S. aureus* LMG 21674. On the contrary, against the strain MRSA, it seems best to use the mixtures. Thymol and *β*-caryophyllene both induced a 4-fold decrease of the concentration of penicillin V. Together (M9), the MIC reached an 8-fold decrease (4 to 0.5 μg/mL). The best indirect activity against the strain MRSA was observed with the mixture M11 (33.3% of each compound), which, in association with penicillin V, decreased the concentration of the antibiotic by 16-fold (4 to 0.25 μg/mL).

## 4. Conclusions

The chemical composition of essential oils is very complex and varied, depending on many factors, both internal and external, to the plant. From our point of view, this high variability could complicate their possible clinical development as effective antibacterial agents. The eight thyme chemotype thymol essential oils purchased on the market all presented a stable chemical composition due to the European pharmacopeia standards or the close geographic origin. By contrast, the chemical composition of ajowan essential oil, which does not have a defined chemotype and is not described in the European pharmacopeia, can be quite varied. In our attempt to standardize the chemical composition by making homemade essential oil mixtures using their major compounds, it appears that in the case of ajowan and thyme essential oils the use of thymol alone as a major compound would be of most benefit. The other two major compounds found in these oils, *p*-cymene and *γ*-terpinene, do not act in synergy with the thymol. Surprisingly, when testing some minor compounds, it was observed that *β*-caryophyllene would be a very good candidate as an antibacterial agent against infections caused by *Staphylococcus aureus*. More studies on the mechanisms of this compound and on its in vivo activities would nevertheless be necessary to further its possible therapeutic development.

## Figures and Tables

**Table 1 antibiotics-10-00584-t001:** Ajowan essential oils (AW) (*Trachyspermum ammi* L.) and thyme chemotype thymol essential oils (TT) (*Thymus vulgaris* L./*Thymus zygis* Loefl. (L.)) purchased from different suppliers.

Sample	Supplier	Name	Batch #	Origin	Organ	€/L
AW-S1	Pranarôm ^1^	*T. ammi* L.	OF36317	India	Fruit	92.20 €
AW-S2	Terraïa ^2^	*T. ammi* L.	191517	India *	Seed	39.50 €
AW-S3	Voshuiles ^3^	*T. ammi* L.	190250	India	Seed	28.50 €
AW-S4	Aromazone ^4^	*T. ammi* L.	20HE0221/1	India *	Seed	29.00 €
AW-S5	Bioflore ^5^	*T. ammi* L.	IL1903	India	Fruit	61.00 €
AW-S6	Floressence ^6^	*T. ammi* L.	EXA5294	India	ns	42.40 €
AW-S7	Botanica ^6^	*T. ammi* L.	ASO101SSF6213	India *	Fruit	46.00 €
AW-S8	Sjankara ^1^	*T. ammi* L.	1008.009	India	Seed	71.18 €
TT-S1	Voshuiles ^3^	*T. vulgaris* L.	200200	Spain	F. top	57.00 €
TT-S2	Voshuiles ^3^	*T. zygis* Loefl. (L.)	B20053	Spain *	A. part	90.00 €
TT-S3	Aromazone ^4^	*T. vulgaris* L.	19HE0236/5	Spain	F. top	55.00 €
TT-S4	Aromazone ^4^	*T. vulgaris* L.	20HE0081/1	France *	F. top	88.00 €
TT-S5	Bioflore ^7^	*T. vulgaris* L.	IL1812	France *	F. top	178.00 €
TT-S6	Botanica ^6^	*T. vulgaris* L.	LSB103116	Spain *	Flower	74.60 €
TT-S7	Valnet ^8^	T. vulgaris L.	18K027E	France *	F. top	157.40 €
TT-S8	Puressentiel ^9^	*T. zygis* Loefl. (L.)	unreadable	France *	A. part	170.00 €

Order place/website: 1–drugstore; 2–onatera.com; 3–voshuiles.com; 4–aroma-zone.com; 5–bioflore.be; 6–lecomptoirdessences.be; 7–kissplanet.shop; 8–docteurvalnet.com; 9–puressentiel.com. F. top—flowering top, A. part—aerial part, ns—not specified. * Organic (AB mark and European label).

**Table 2 antibiotics-10-00584-t002:** Chemical composition (main compounds) of the eight thyme chemotype thymol essential oils (*n* = 3, percentage ± SD).

Sample	Thymol	*p*-cymene	*γ*-terpinene	Carvacrol	Linalool	*β*-myrcene	*β*-caryophyllene
TT-S1	63.86 ± 1.11	20.92 ± 0.38	6.64 ± 0.38	2.71 ± 0.44	0.90 ± 0.11	0.39 ± 0.03	0.56 ± 0.04
TT-S2	56.04 ± 0.44	22.92 ± 0.32	11.55 ± 0.28	5.04 ± 0.08	0.76 ± 0.05	0.30 ± 0.01	0.24 ± 0.02
TT-S3	54.56 ± 0.44	26.20 ± 0.37	9.55 ± 0.13	3.53 ± 0.14	1.29 ± 0.06	0.21 ± 0.02	0.93 ± 0.06
TT-S4	58.09 ± 1.35	23.71 ± 2.55	8.80 ± 0.12	3.35 ± 0.10	1.60 ± 0.29	0.38 ± 0.06	0.59 ± 0.23
TT-S5	57.38 ± 1.07	24.51 ± 4.00	7.16 ± 0.08	3.40 ± 0.26	2.45 ± 0.52	0.25 ± 0.09	0.93 ± 0.43
TT-S6	55.70 ± 0.85	28.52 ± 0.78	8.98 ± 0.15	2.34 ± 0.19	1.39 ± 0.05	0.28 ± 0.02	0.24 ± 0.01
TT-S7	48.64 ± 0.64	32.17 ± 0.91	10.23 ± 0.07	1.73 ± 0.16	0.87 ± 0.04	0.43 ± 0.02	1.21 ± 0.05
TT-S8	61.05 ± 0.09	26.69 ± 1.30	3.42 ± 0.11	2.70 ± 0.17	1.84 ± 0.09	0.19 ± 0.01	0.59 ± 0.05

**Table 3 antibiotics-10-00584-t003:** Chemical composition (main compounds) of the eight ajowan essential oils (*n* = 3, percentage ± SD).

Sample	Thymol	*γ*-terpinene	*p*-cymene	*β*-pinene
AW-S1	53.66 ± 0.90	15.77 ± 0.44	26.36 ± 0.68	2.04 ± 0.40
AW-S2	33.69 ± 0.59	43.77 ± 1.28	19.43 ± 0.34	0.97 ± 0.31
AW-S3	32.41 ± 0.62	44.17 ± 1.20	21.46 ± 0.81	0.83 ± 0.25
AW-S4	41.26 ± 0.98	35.15 ± 0.86	20.05 ± 0.89	1.27 ± 0.36
AW-S5	35.57 ± 1.97	27.76 ± 0.37	36.15 ± 2.43	0.00 ± 0.00
AW-S6	58.12 ± 1.40	19.67 ± 0.77	19.11 ± 1.97	0.54 ± 0.16
AW-S7	28.21 ± 0.90	24.17 ± 0.65	44.00 ± 1.43	0.32 ± 0.09
AW-S8	35.62 ± 0.70	25.21 ± 0.69	38.64 ± 1.31	0.00 ± 0.00

**Table 4 antibiotics-10-00584-t004:**
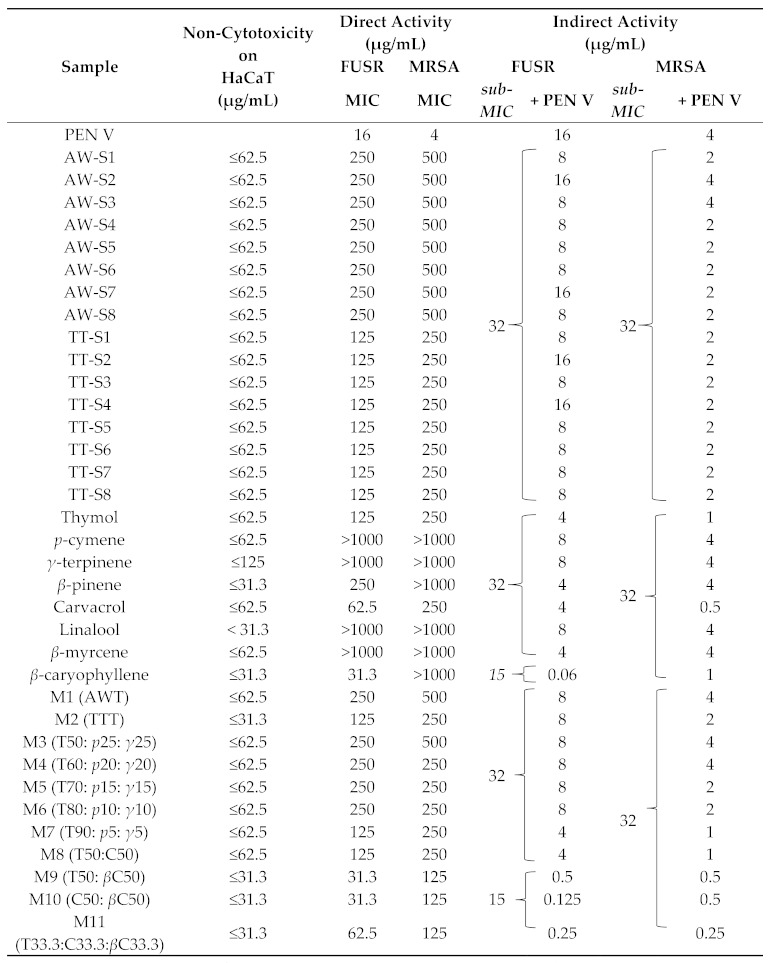
Biological activities of ajowan and thyme chemotype thymol essential oils (AW-S1 to S8 and TT-S1 to S8), the chemical compounds and the mixtures (M1 to M11). The cytotoxic activity was assessed on the keratinocyte cell line (HaCaT) and direct activity and indirect (in association with the penicillin V (PEN V)) antibacterial activity were assessed against two Gram-positive bacteria, *Staphylococcus aureus* LMG 21,674 (FUSR) and LMG 15,875 (MRSA).

AB—antibiotic, AWT—mixture type ajowan (50% thymol, 25% *γ*-terpinene, 20% *p*-cymene, and 5% *β*-pinene), TTT—mixture type thymol thyme (50% thymol, 20% *p*-cymene, 10% *γ*-terpinene, 5% *β*-myrcene, 5% linalool, 5% *β*-caryophyllene, and 5% carvacrol), TX—X percentage of thymol, *p*X—X percentage of *p*-cymene, *γ*X—X percentage of *γ*-terpinene, CX—X percentage of carvacrol, *β*CX—X percentage of *β*-caryophyllene.

## Data Availability

The data used to support the findings of this study are included in this study.

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
