# Peer review of "Antibacterial Activities of Homemade Matrices Miming Essential Oils Compared to Commercial Ones"

_antibiotics, 2021, doi:10.3390/antibiotics10050584_

Round 1
Reviewer 1 Report
Minor remarks
I suggest changing the title of the manuscript because “vs” should be omitted.
“et al.” should be presented in italics.
Avoid the first-person plural. Use only the third-person singular in the scientific paper.
Delete a blank space between quantity and percentage through the text.
All other minor remarks are presented in the document.
Major remarks
Avoid lumping the references. Each reference should be discussed separately.
It is desirable to depict the GC-MS chromatograms as the supplement.

Reviewer 2 Report
It is a very interesting study, congratulations to the authors. I have only one suggestion - for me intrinsic/outrinsic sounds really strange (in the Introduction and in the Conclusion part). We use internal/external, or in the Introduction part biotic/abiotic factors can be used.
Reviewer 3 Report
Your paper addresses a topical medical issue, because finding herbal remedies to resolve drug resistance is currently a major research topic (but also a challenge) in contemporary medicine, pharmacy, biochemistry and chemistry.
The Abstract is clear and relevant. However, I recommend to insert the name of the species at R 28-29 (Ajowan is Trachyspermum ammi L., Thyme is …).
The research is well motivated in Introduction,
In general, the research is well conducted. However:
- I recommend that in point 2.5. to insert a justification regarding the selection of the concentration range used to establish the MIC.
- I consider that the sentence from R128-130 (,,The tested strains ….to fusidic acid)”) must be corrected (MRSA / FUSR ?)
The results are clearly presented. The discutions are complex and relevant. The bibliography is exhaustive and correctly written.
